# Serological Evidence of Exposure to Eurasian-Lineage HPAI H5N1 Clade 2.3.4.4b in Wild Mammals in Ohio, USA, 2024–2025

**DOI:** 10.3390/v17101388

**Published:** 2025-10-18

**Authors:** Mohammad Jawad Jahid, Madison C. Owsiany, Lauren M. Smith, Bryant M. Foreman, Zijing Cao, Deborah L. Carter, David E. Stallknecht, Brendan Shirkey, Rebecca L. Poulson, Jacqueline M. Nolting

**Affiliations:** 1Department of Veterinary Preventive Medicine, College of Veterinary Medicine, The Ohio State University, Columbus, OH 43210, USA; jahid.1@osu.edu (M.J.J.); owsiany.2@osu.edu (M.C.O.); smith.9701@osu.edu (L.M.S.); foreman.183@osu.edu (B.M.F.); 2Southeastern Cooperative Wildlife Disease Study, Department of Population Health, College of Veterinary Medicine, University of Georgia, Athens, GA 30602, USA; Zijing.Cao@uga.edu (Z.C.); dlcarter@uga.edu (D.L.C.); dstall@uga.edu (D.E.S.); rpoulson@uga.edu (R.L.P.); 3Winous Point Marsh Conservancy, Port Clinton, OH 43452, USA; brendan@winous.org

**Keywords:** highly pathogenic avian influenza H5N1 2.3.4.4b, wildlife, surveillance, influenza ecology, influenza epidemiology

## Abstract

The Goose/Guandong lineage of highly pathogenic avian influenza virus [A/Goose/Guangdong/1/1996(H5N1)] is the progenitor of the currently circulating Eurasian-lineage highly pathogenic avian influenza H5N1 clade 2.3.4.4b and has been the most consequential highly pathogenic avian influenza lineage globally. Despite increased reports of infections, the extent of exposure and role of wild mammals in the ecology and transmission dynamics of the virus remains poorly understood. We surveyed wild mammals in Ohio, United States to investigate the potential spillover of highly pathogenic H5N1 avian influenza clade 2.3.4.4b. While no active infections—defined as positive results indicative of viral replication and potential propagation—were detected by swab-based molecular tests, serological assays revealed antibodies against multiple avian influenza virus antigens in raccoons and opossums. Specifically, antibodies to avian influenza virus nucleoprotein were detected in 54.9% (*n* = 61) of samples using enzyme-linked immunosorbent assay; antibodies to Eurasian-lineage highly pathogenic avian influenza H5 clade 2.3.4.4b and North American low pathogenic avian influenza H5 were detected in 43.2% (*n* = 48) and 22.5% (*n* = 25) of samples, respectively, using virus neutralization assays; and antibodies to avian influenza virus neuraminidase were detected in 44.1% (*n* = 49) of samples using enzyme-linked lectin assay. All seropositive animals were sampled at Ohio marshes with previously confirmed highly pathogenic avian influenza H5N1 detections in waterfowl. These findings suggest prior exposure of wild mammals to these viruses without mortality events. Wild mammals may play an intermediary role in the mammalian adaptation of avian influenza A viruses. Therefore, ongoing surveillance of wild mammals is crucial for assessing the risk to public health.

## 1. Introduction

In nature, influenza A viruses (IAVs) infect a broad range of bird species [1]; however, they have also shown the ability to infect numerous mammals, including the order of Carnivora [2,3]. Spread of IAVs from avian reservoirs to non-human mammalian hosts were initially reported by isolation of avian H7N7 [1] and H4N5 [4] from harbor seals, H13N2 and H13N9 from pilot whales [5], and H10N4 from farmed American mink [6,7]. In these species, IAVs caused severe illness and/or death. IAV infection was also documented in carnivore species, including dogs [8], cats [9], and captive bears [10], which experienced asymptomatic or mild clinical symptoms. Meanwhile, highly pathogenic avian influenza (HPAI) A(H5N1) viruses caused systemic infection and/or death in numerous carnivore species, including tigers and leopards [11], domestic cats [9,12,13], palm civets [14], domestic dogs [15], and stone martens [16]. Dwelling in aquatic habitats and feeding and/or scavenging on infected wild birds were the presumable source of these mammalian infections. Wild carnivore species that opportunistically feed on wild birds may, therefore, be at a higher risk of infection with HPAI viruses [2,17].

HPAI viruses of the Goose/Guandong (Gs/GD) lineage [A/Goose/Guangdong/1/1996(H5N1)], first detected in 1996 from a farmed goose in Guangdong Province, China, have been globally the most consequential HPAI lineage to date [18,19]. They are the ancestors of the currently circulating HPAI H5N1 2.3.4.4b, which were detected in North America in December 2021 [20,21], and have demonstrated the capacity to spread widely across the Americas, affecting wild birds, poultry, wildlife, livestock, and humans [22,23,24,25,26]. Of the wild mammalian species, these viruses have predominantly been detected in red fox, striped skunk, mountain lion, raccoon, harbor seal, and bobcat [27]. Infections have been dead-end and are mostly linked to direct contact and/or preying and scavenging on infected birds [28]. The growing number of cases in mammals [27], combined with the evidence of lateral transmission of HPAI H5N1 in marine mammals in Chile and Peru [29] and farmed American mink in Spain [30], raise concerns as they involved PB2 mammalian-adaptation mutations, which underscore the virus’s potential to adapt to humans [31].

Despite the increasing reports of the Eurasian (EA)-lineage HPAI H5 2.3.4.4b (EA HP H5, hereafter) infections in a wide range of wild and domestic animals, the extent of exposure and potential role of wild mammal species in the ecology and transmission of the virus remains poorly understood. Limited active molecular and serological surveillance has hindered our understanding of the extent of infection and exposure in these species. Surveillance of wild mammals at the avian–mammalian interface was needed to address this gap, as well as to assess interspecies transmission risks, improving early warning systems for panzootic and pandemic threats [28,32]. In this study, we conducted molecular and serological surveillance of wild mammals in Ohio, USA to evaluate evidence of active infection and/or prior exposure to EA HP H5 viruses. Given that infections of severe acute respiratory syndrome Coronavirus-2 (SARS-CoV-2) viruses were previously reported in wild mammals in Ohio [33], we also opportunistically screened samples collected in this study for SARS-CoV-2. These findings may offer insights into species-specific exposure factors and ecological drivers of the spillover, providing essential data for future surveillance and risk assessment efforts at the wild bird–mammal interface.

## 2. Materials and Methods

Active, opportunistic IAV surveillance was conducted in wild mammals in Ohio, USA from March 2024 to April 2025. The study was approved by The Ohio State University, Institutional Animal Care and Use Committee, protocol 2024A00000074. Study sites were selected based on their accessibility to wild mammals, habitat composition, and/or prior reports of highly pathogenic avian influenza (HPAI) H5N1 detections in wild waterfowl. Nasal or nasopharyngeal swabs and blood samples were opportunistically collected from a range of individual wild mammalian species. Specimens were obtained from live-trapped animals in an Ohio marsh (*n* = 1 marsh) in Ottawa county [27] with known waterfowl HPAI circulation and from animals admitted to wildlife rehabilitation centers (*n* = 2 centers) located in both rural and urban areas of Ohio. Sampling was conducted by wildlife biologists at marsh habitats and by licensed veterinarians at the wildlife rehabilitation centers. Live-trapped animals were humanely euthanized before sample collection as part of the wildlife conservatory program’s efforts to prevent waterfowl predation (https://ohiodnr.gov/discover-and-learn/land-water/issues-for-landowners/nuisance-raccoons, accessed 28 June 2025) and rehab animals were sampled during veterinary inspection at intake. Animals were sampled only once across all the sampling sites.

Nasopharyngeal swab samples were collected from small mammals at rehab centers using sterile nasopharyngeal calcium alginate-tipped applicators (catalog # 25800 A50, Fisher Scientific, Hampton, NH, USA) and nasal swabs were collected from larger animals trapped in the marsh using sterile polyester-tipped applicators (catalog # 25806 2PD, Fisher Scientific, Hampton, NH, USA). Swabs were immediately placed in virus transport medium containing brain–heart infusion broth (BHIB) which contained penicillin (10,000 units/mL) and streptomycin (10 mg/mL) [34], and transported on dry ice or liquid nitrogen to the lab where they were stored at −80 °C until molecular analysis was performed. Blood samples were collected in serum separator tubes (BD SST blood collection tubes; catalog # 367989, Benton-Dickson, Franklin, NJ, USA), allowed to clot, and centrifuged at 1200× *g* for 10 min; and sera was removed and stored at −20 °C in cryovials for serologic analyses.

Total RNA was extracted from the BHIB by the MagMAX™ Viral/Pathogen Nucleic Acid Isolation Kit (catalog # A48383, Fisher Scientific, Hampton, NH, USA) using the KingFisher Flex Purification System w/ 96 Deep Well PCR Magmax (catalog # 50-152-7922, Fisher Scientific, Hampton, NH, USA) according to the manufacturers’ instructions. The presence of influenza A virus (IAV) and severe acute respiratory syndrome-2 (SARS-CoV-2) RNA was assessed using real-time reverse transcription PCR (rRT-PCR) with the VetMAX™-Gold SIV Detection Kit (Catalog # 4415200, Fisher Scientific, Hampton, NH, USA) and the TaqPath^TM^ COVID-19 RT-PCR Kit (catalog # 21-200-329, Fisher Scientific, Hampton, NH, USA), respectively, according to the manufacturers’ protocols. Samples with a cycle threshold (Ct) value of ≤40 were recorded as positive. Any samples that tested positive for IAV were subsequently screened for the H5 using primers and probe set was designed to target the hemagglutinin gene of Goose/Guangdong (Gs/GD) lineage IAV subtype H5, clade 2.3.4.4 (NVSL-WI-1732, National Veterinary Service Laboratory, Ames, IA, USA).

Serological analysis for detection of antibodies to IAVs were conducted using methods described previously [31,35]. Briefly, a commercially available MultiS-Screen Ab blocking ELISA (catalog # 99-12119; IDEXX Laboratories, Westbrook, ME, USA) was used for the detection of antibodies to IAV nucleoprotein (NP), with all the bELISA-positive sera further screened for antibodies to H5 and to N1 using virus neutralization (VN) and an enzyme-linked lectin assay (ELLA), respectively. Two reverse genetic (RG) antigens were used in VN assays to screen for antibodies to EA HP H5 and to North American (NAm.) low pathogenic avian influenza (LPAI) H5. The antigens included IDCDC-RG71A, which incorporates Eurasian-lineage hemagglutinin (HA) and neuraminidase (NA) genes from A/Astrakhan/3212/2020 (H5N8) on an A/Puerto Rico/8/1934 (H1N1) (PR8) backbone, and LP-RGBWT/TX, which contains North American HA and NA genes from A/blue-winged teal/AI12–4150/Texas/2012 (H5N2) on a PR8 backbone [31]. The HA of IDCDC-RG71A was modified at the protease cleavage site to confer a low pathogenic influenza A virus phenotype. For the ELLA, A/ruddy turnstone/New Jersey/AI13–2948/2013 (H10N1) served as the antigen [31]. Conservative positive threshold titers were used to determine seropositivity: bELISA Serum/Negative (S/N) <0.7; H5 VN ≥20; and N1 ELLA, ≥80.

Geometric mean titers (GMTs) were calculated as the antilogarithm of the arithmetic means of log2-transformed titers [36]. The Mann–Whitney U test was used to evaluate the statistical significance of differences in overall GMTs between the Eurasian HP H5 and NaAM LPAI H5 groups, as well as within-month variations in GMTs between the two groups, based on the raw titer observations. The test was two-tailed and *p*-values of <0.05 were considered as significant. Descriptive analyses were performed using STATA BE V. 18.5, with figures generated in GraphPad Prism V. 10.5.0.

## 3. Results

A total of 292 swab samples were collected from sixteen individual wild mammal species; 111 serum samples were collected from eight of these species (Table 1). All the raccoon samples and a subset of opossum samples (*n* = 3 swab and *n* = 2 serum) were collected from live-trapped animals at Ohio marsh habitats. The remaining samples were collected from animals admitted to wildlife rehabilitation centers in Ohio. None of the swab samples tested positive for either IAV or SARS-CoV-2 viral RNA. However, seropositivity to IAV was detected among the tested serum samples (Table 1).

Specifically, antibodies were detected against IAV nucleoprotein (NP) in 54.9% (*n* = 61) of the serum samples; EA HP H5 in 43.2% (*n* = 48); NAm LPAI H5 in 22.5% (*n* = 25); and IAV neuraminidase (N1) in 44.1% (*n* = 49) (Table 2).

All the seropositive results were observed exclusively in raccoons and a single opossum sampled at Ohio marsh habitats (Figure 1). By contrast, no seropositive results were observed among animals sampled from wildlife rehabilitation centers.

Additionally, the geometric mean titer (GMT) for EA HP H5 was higher (99.47) than for NAm. LPAI H5 (17.76) in the tested serum samples, and this observation was statistically significant (*p*-value <0.001). Throughout the 2024–2025 sampling period, except for August 2024, the GMTs for EA HP H5 remained higher than that of NAm. LPAI H5, ranging from 10 to 341 and from 10 to 40, respectively (Figure 2).

## 4. Discussion

This study provides serological evidence of EA HP H5 exposure in raccoons and opossums in Ohio, USA during a period of known regional HPAI H5N1 circulation in numerous animal species [22,26,30,32,37,38]. The EA HP H5 seropositive animals were also seropositive to IAV N1, indicating their exposure to EA HP H5N1 viruses. While antigenic cross-reactivity to LPAI H5N1 viruses cannot be fully excluded, the detection of EA HP H5-specific antibodies is supported by the higher GMT observed, suggesting a closer antigenic match to these viruses [35]. This observation is also consistent with the higher detections of EA HP H5 in wild mammals since the start of the outbreak in North America [39].

Interestingly, all the raccoons and the seropositive opossum were sampled at Ohio marshes with known HPAI H5N1 detections in waterfowl, e.g., Ottawa county, Ohio [39]. In this region, the highest waterfowl HPAI H5N1 detections occurred in autumn of 2023, followed by limited or no detections in the subsequent seasons through spring of 2025 [39]. Despite this apparent decline in reported detections, our study revealed high GMTs to EA HP H5 among wild mammals during spring of 2024 and again in winter of 2024 and spring of 2025 (Figure 2). These findings suggest that wild mammals were likely exposed to EA HP H5 during the migratory season in autumn of 2023, with this exposure contributing to the elevated GMTs detected in spring of 2024. The marked decrease in the GMTs observed during summer of 2024 coincided with the reduced prevalence of IAVs and EA HP H5 in waterfowl reported in the area during spring and summer of 2024 [39], indicating limited ongoing exposure of wild mammals during seasons in which migratory waterfowl are not moving south. While waterfowl EA HP H5 detections were not reported by USDA during winter of 2024 (December 2024–February 2025) in this area [39], our current active surveillance in waterfowls in this region (The Ohio State University, Columbus, Ohio, wild-bird IAV surveillance, 2022–2025) detected an overall IAV prevalence of 18% and EA HP H5 prevalence of 6% in waterfowl. These results confirm continued circulation of IAVs in local waterfowl populations and suggest that wild mammals were exposed during this period, potentially explaining the elevated GMTs observed in winter of 2024 and spring of 2025 (Figure 2).

EA HP H5 seropositivity of wild mammals sampled in habitats with known HPAI H5N1 detections in wild birds have also been reported in previous studies [3,35]. This is biologically relevant to the ecological behaviors of mesopredators—high mobility, frequent foraging in aquatic environments, predation, and scavenging on birds, which may lead to repeated exposure through multiple sources, such as predation, scavenging, contaminated water, etc. [40]. Potential for repeated exposure, likely through multiple sources, may also explain the reason for the elevated GMTs to EA HP H5 observed in the current study, particularly during winter of 2024 and spring of 2025. Ecological routes of IAV transmission to wild mammals, e.g., through infected bird carcasses [2] and contaminated water [41], have also been documented experimentally. Further, as part of our active surveillance of IAVs in wild birds inhabiting Ohio marshes, we opportunistically collected samples from live-trapped waterfowl during banding activities (https://ohiodnr.gov, accessed 28 June 2025) (The Ohio State University, Columbus, OH, USA, wild-bird IAV surveillance, 2022–2025). Often, waterfowl within the traps were scavenged by raccoons, suggesting a potential interface between wild birds and mammals that can facilitate virus spread/spillover [17,40,41,42]. The lack of IAV seropositivity in other wild mammal species sampled may be explained by their geographical presence, primarily in urban areas and not in aquatic habitats, feeding preferences, or limited sample size representations.

Absence of IAV RNA in swab samples in the seroconverted raccoons and opossum was expected, as seroconversion to IAV occurs several days after the initial infection, lasting for an unknown duration [43,44]. Thus, by the time sampling occurred, the period of active viral shedding had likely already elapsed [17]. Alternatively, the lack of detectable IAV RNA in swab samples could be due to a transient and brief window of IAV nasal shedding in wild mammals, which has been documented in experimentally infected peridomestic wildlife with EA HP H5 [43]. This possibility highlights inherent limitations of sampling relative to the temporal dynamics of viral shedding [17,45]. Another plausible explanation to this could be the swab sampling method, as respiratory swabs have been shown to be less sensitive than brain tissue samples in the detection of HPAI H5N1 from wildlife species [17], emphasizing the importance of brain tissue samples in wildlife surveillance efforts. This approach is particularly relevant when surveillance is integrated with conservation or nuisance animal removal efforts.

Although SARS-CoV-2 infections have been documented in numerous wildlife species [46,47,48,49,50,51,52,53], no viral detections were recovered in this surveillance effort. This is consistent with the findings of Ehrlich et al., [45] and does not necessarily indicate that these species are not susceptible to SARS-CoV-2. Since we may have missed the active virus shedding window [45], future studies, covering multiple seasons while also incorporating serological surveillance, are warranted to better assess the extent of exposure and susceptibility of these wildlife species to SARS-CoV-2.

Together, these findings emphasize the importance of the continued surveillance of wild mammals, particularly mesocarnivores, at the wildlife–livestock–human interface. Such efforts are essential for understanding the ecological dynamics of IAV spillover and assessing the potential for mammalian adaptation, especially given the increasing number of mammalian HPAI detections in the U.S. and globally.

This study is geographically limited to Ohio, with some under-represented mammalian species; hence, the results may not be fully generalizable to other regions with differing habitats and wild species compositions. Calcium alginate swabs have been reported to inhibit PCR detections of IAVs [54]; therefore, it represents a limitation of this study. Nevertheless, these swabs were the only type available for sampling small mammals at wildlife rehabilitation centers. Also, our study is limited in that it lacks age records for the seropositive raccoons, restricting assessment of age-related patterns in infection or immunity [40]. Therefore, future surveillance efforts should aim to address these, by (a) expanding sampling efforts both temporally—across multiple seasons—and geographically, covering a broader range of habitats and regions; (b) incorporating more epidemiologically informative data like age; (c) increasing the representation of under-sampled mammalian species; and (d) including brain tissue samples when applicable.

## Figures and Tables

**Figure 1 viruses-17-01388-f001:**
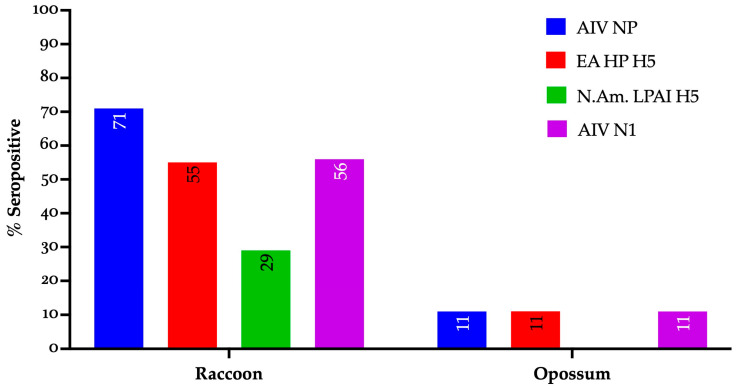
Seropositivity profile of individual wild mammals to IAV gene segments and lineages (EA and NAm.) sampled in Ohio during 2024–2025. AIV NP: avian influenza nucleoprotein, EA HP H5: Eurasian-lineage highly pathogenic avian influenza H5 clade 2.3.4.4b, N.Am. LPAI H5: North American low pathogenic avian influenza H5, AIV N1: avian influenza virus neuraminidase subtype 1.

**Figure 2 viruses-17-01388-f002:**
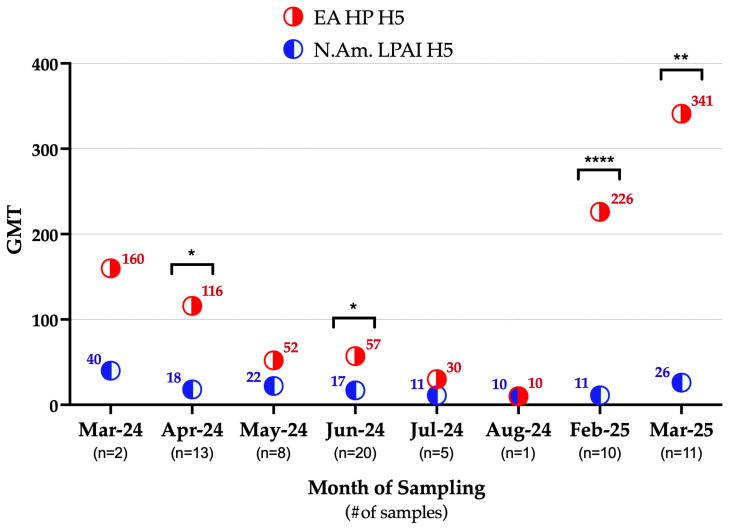
Temporal trend in geometric mean titers (GMTs) of the EA HP H5 and NAm. LPAI H5 in wild mammals sampled in Ohio during 2024–2025. Within the month, statistically significant observations are denoted by asterisks (* *p* = 0.0125, 0.0228; ** *p* = 0.0041; **** *p* < 0.001). EA HP H5: highly pathogenic avian influenza H5 clade 2.3.4.4b, N.Am. LPAI H5: North American low pathogenic avian influenza H5, GMT: geometric mean titer.

**Table 1 viruses-17-01388-t001:** Counts of swab and serum samples collected from wild mammals in Ohio (2024–2025), with number of seropositive animals to IAV NP by sampling sites.

*Species* *(Common Name)*	*Species* *(Scientific Name)*	*# of Swab Samples*	*# of Serum Samples*	*# IAV Seropositive/# of Samples (%)*	
				Marshes	Rehabilitation Centers
American beaver	*Castor canadensis*	2	0	-	-
American mink	*Mustela vison*	3	0	-	-
Big brown bat	*Eptesicus fuscus*	4	0	-	-
Chipmunk	*Tamias striatus*	2	2	-	0/2 (2)
Eastern cottontail	*Sylvilagus floridanus*	7	2	-	0/2 (2)
Fox squirrel	*Sciurus niger*	3	0	-	-
Gray squirrel	*Sciurus carolinensisn*	10	10	-	0/10 (10)
Groundhog	*Marmota monax*	2	1	-	0/1 (0)
Muskrat	*Ondatra zibethicus*	98	0	-	-
River otter	*Lontra canadensis*	1	0	-	-
Raccoon	*Procyon lotor*	142	85	60/85 (70.6%)	-
Red squirrel	*Tamiasciurus hudsonicus*	1	1	-	0/1 (0)
Stripped skunk	*Mephitis mephitis*	1	0	-	-
Opossum	*Didelphis virginiana*	13	9	1/2 (50%)	0/7 (0)
White-footed mouse	*Peromyscus leucopus*	1	1	-	0/1 (0)
Woodchuck	*Marmota monax*	2	0	-	-
Total		292	111	61/87 (70.1%)	0/24 (0%)
*Grand Total*				*61/111 (54.9%)*	

**Table 2 viruses-17-01388-t002:** Comparison of antibody titer distributions across VN (EA HP H5, NAm. LPAI H5) and ELLA (N1) assays.

TITER GROUP	VN (EA HP H5) COUNT (%)	VN (NAM. LPAI H5) COUNT (%)	ELLA (N1) COUNT (%)
<20	21 (30)	44 (62.8)	14 (19.7)
20	3 (4.3)	11 (15.7)	7 (9.8)
40	3 (4.3)	7 (10)	1 (1.4)
80	9 (12.8)	0 (0)	1 (1.4)
160	9 (12.8)	7 (10)	2 (2.8)
320	7 (10)	1 (1.4)	2 (2.8)
640	6 (8.6)	0 (0)	9 (12.7)
1280	7 (10)	0 (0)	9 (12.7)
2560	5 (7.1)	0 (0)	26 (36.6)
TOTAL	70 (100)	70 (100)	71 (100)

VN: virus neutralization, ELLA: enzyme-linked lectin assay, EA HP H5: Eurasian-lineage highly pathogenic avian influenza A(H5) 2.3.4.4b, NAm. LPAI H5: North American-lineage low pathogenic avian influenza A(H5).

## Data Availability

All the data are contained within this article.

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
