# Peer review of "Serological Evidence of Exposure to Eurasian-Lineage HPAI H5N1 Clade 2.3.4.4b in Wild Mammals in Ohio, USA, 2024–2025"

_viruses, 2025, doi:10.3390/v17101388_

Round 1

Reviewer 1 Report

Comments and Suggestions for Authors

In their manuscript, “Serological Evidence of Exposure to Eurasian-lineage HPAI H5N1 clade 2.3.4.4b in Wild Mammals in Ohio, USA, 2024–2025”, Jahid et al present their findings from surveillance for H5N1 in wild mammals in Ohio. Their serological findings indicate that raccoons and opossums may have been exposed to H5N1. The study is important as it adds to our understanding of the risk of HPAI to various mammalian species but can be improved.

Major comments:

  • More clarity is needed when describing the sampling procedures:
    1. A map showing the locations of the marshes/shelters showing the number of samples obtained from each would be helpful.
    2. The cat no. for swabs used from puritan are not correct as listed ex. 25-806 2PD
    3. Were naso-pharyngeal swabs really obtained or were they nasal swabs? Obtaining naso-pharyngeal swabs from small mammals is tricky and requires sedation.
    4. The tip of the smaller size swab is of calcium alginate which could reduce the yield of DNA and interfere with PCR. This should be discussed.
    5. Were individual shelter animals sampled more than once?
    6. How was sample size determined or was posthoc power analysis conducted especially that swabs were negative
    7. How were sampling sites selected?
  • Table 1 and the results showing the number of samples obtained should be expanded to show the type of site the samples were obtained from (marsh/shelter).
  • A more detailed presentation of the serology data should be presented in a table ex. summarizing titers by categories (<20, 20, 40, 80, ….) and the percent with each category.
  • Statistical analysis for GMT comparisons is needed for both overall and within month.
  • In the discussion, it is essential to better describe and add a lot more detail on how antibody detection was following virus detection in wild birds at each sampling site. Additionally, figure 2 appears to show a seasonal pattern with antibodies dropping over the summer months, this should also be discussed.
  • In the discussion (line 183), more ecological information on animal behavior, especially raccoons and opossums in aquatic habitats would be important to give readers a clearer understanding of the interface.
  • Lack of PCR detection is likely due to sample size or restricted number of sampling sites.

Minor comments:

  • In the abstract, refrain from using abbreviations especially in the sentence showing the serological assays as it is confusing and rather spell out the assay used as indicated later in the methods ex. NP ELISA was used to screen sera and positives were further tested by VN for H5 and ELLA for N1.
  • Abstract line 28: Delete ‘and recovery’
  • In the M&M: List the viruses used in the various assays; the HA and NA of the rg virus and the backbone used, the LPAI virus, and the virus used for ELLA
  • In figure 2, it would be helpful to show the GMT value over the dots

Author Response

Major comments:

  • More clarity is needed when describing the sampling procedures:
    1. A map showing the locations of the marshes/shelters showing the number of samples obtained from each would be helpful.

The authors presented this data in Table 1. and in methods’ section, line #s 85-88. 

    1. The cat no. for swabs used from puritan are not correct as listed ex. 25-806 2PD

This catalog # is accessible through Fisher. I have updated the link to https://www.fishersci.com/ in line# 92.

    1. Were naso-pharyngeal swabs really obtained or were they nasal swabs? Obtaining naso-pharyngeal swabs from small mammals is tricky and requires sedation.

Nasal swabs were collected from animals at marshes, whereas nasopharyngeal swabs were collected from rehabilitation centers where sick and/or death animals were sampled. This is addressed in line #s 85-88.

    1. The tip of the smaller size swab is of calcium alginate which could reduce the yield of DNA and interfere with PCR. This should be discussed.

Calcium alginate swabs were not used in this study. I removed this piece!

    1. Were individual shelter animals sampled more than once?

No, each animal was only tested once

    1. How was sample size determined or was posthoc power analysis conducted especially that swabs were negative

Samples were collected opportunistically in combination with wildlife conservation and rehabilitation efforts. This is addressed in line #s 87-90.

    1. How were sampling sites selected?

Please refer to line #s 85-88.

  • Table 1 and the results showing the number of samples obtained should be expanded to show the type of site the samples were obtained from (marsh/shelter).

Done!

  • A more detailed presentation of the serology data should be presented in a table ex. summarizing titers by categories (<20, 20, 40, 80, ….) and the percent with each category.

Added as a Supplementary table, e.g., supplementary 1.

  • Statistical analysis for GMT comparisons is needed for both overall and within month.

Please refer to line #s 162-165 and figure 2.

  • In the discussion, it is essential to better describe and add a lot more detail on how antibody detection was following virus detection in wild birds at each sampling site. Additionally, figure 2 appears to show a seasonal pattern with antibodies dropping over the summer months, this should also be discussed.

Done, some more information added! Please refer to line #s 182-196.

  • In the discussion (line 183), more ecological information on animal behavior, especially raccoons and opossums in aquatic habitats would be important to give readers a clearer understanding of the interface.

Please refer to line #s 189-204.

  • Lack of PCR detection is likely due to sample size or restricted number of sampling sites.

Yes! Please refer to line #s 209-220 supporting your points as well as other alternative possibilities.

Minor comments:

  • In the abstract, refrain from using abbreviations especially in the sentence showing the serological assays as it is confusing and rather spell out the assay used as indicated later in the methods ex. NP ELISA was used to screen sera and positives were further tested by VN for H5 and ELLA for N1.

Edited to remove abbreviations

  • Abstract line 28: Delete ‘and recovery’

That’s an important piece of our findings, indicating that EA HP H5 was not lethal in these seropositive animals, as opposed to other mammalian species, e.g., cats and ferrets.

  • In the M&M: List the viruses used in the various assays; the HA and NA of the rg virus and the backbone used, the LPAI virus, and the virus used for ELLA

Done, please refer to line #s 130-137.

  • In figure 2, it would be helpful to show the GMT value over the dots

Done!

Reviewer 2 Report

Comments and Suggestions for Authors

This manuscript is about serological evidence of EA HP H5N1 (clade 2.3.4.4b) in racoons and opossum. Considering worldwide concerns about H5N1-related interspecies transmission, this manuscript provides timely information on its potential transmission among wild mammals. Although this study is interesting, there are several points to be considered to improve the manuscript.

- While introduction and discussion are well written ‘results’ section needs to be improved to provide more details and information.

- The serological tests were done with bELISA against NP, VN against RG viruses (HA and NA from H5N1 and H5N2 with PR8 backbone), and N1 ELLA. The data could be more convincing if HI assay and N2-specific ELLA data. This reviewer thinks those additional data can provide more information whether the seropositives were due to EA HP H5N1 or NAm. LPAI H5.

- It is a little bit confusing to interpret the result description (line 138-143) and figure 1. The numeric data in figure 1 seems not matched with descripted one in the paragraph.

- In figure 2, Error bar would be helpful with statistical analysis for their significances.

<minor>

- in table 1, at second column, Specie -> Species

Author Response

This manuscript is about serological evidence of EA HP H5N1 (clade 2.3.4.4b) in racoons and opossum. Considering worldwide concerns about H5N1-related interspecies transmission, this manuscript provides timely information on its potential transmission among wild mammals. Although this study is interesting, there are several points to be considered to improve the manuscript.

- While introduction and discussion are well written ‘results’ section needs to be improved to provide more details and information.

- The serological tests were done with bELISA against NP, VN against RG viruses (HA and NA from H5N1 and H5N2 with PR8 backbone), and N1 ELLA. The data could be more convincing if HI assay and N2-specific ELLA data. This reviewer thinks those additional data can provide more information whether the seropositives were due to EA HP H5N1 or NAm. LPAI H5.

We didn’t perform HI in this study. Whether or not the seropositivity were due to EA HP H5 or NAm. LPAI H5 are discussed in line #s 174-181.

- It is a little bit confusing to interpret the result description (line 138-143) and figure 1. The numeric data in figure 1 seems not matched with descripted one in the paragraph.

The denominator for statistics in line #s 153-155 are all animals that were tested (e.g., 111). While the denominator for line # 155-157 and figure 1 are only in the seropositive animals (e.g., only raccoons and opossum).

- In figure 2, Error bar would be helpful with statistical analysis for their significances.

A test of statistical significance has not been done for these GMT observations. The caveat, e.g., cross-reactivity, for the observed GMTs and the assumptions to shadow that is discussed in line #s 177-181.

<minor>

- in table 1, at second column, Specie -> Species

Solved!

Reviewer 3 Report

Comments and Suggestions for Authors

In this manuscript, The authors performed a survey on wild mammals in Ohio, United States to investigate potential spillover of EA HP H5. These findings suggest prior exposure and recovery of wild mammals to these viruses. Wild mammals may play an intermediary role in the mammalian adaptation of avian influenza A viruses. But, there are some matters that need to be clarified.

Manuscript:

Point 1: Line 22: what's meaning of active infection, please describe this issue in detail.

Point 2: Line 133: In results part, None of the swab samples tested positive for either AIV or SARS-CoV-2 viral RNA. Why was the virus only detected in the serum sample but not in the swab? The serological test might have resulted in a false positive.

Point 3: Line 44: Which branch does the H5N1 detected in this paper belong to? And how is it different and related to the H5N1 that infected dairy cows at American dairy farms in 2023?

Comments on the Quality of English Language

The language and logic of the English papers need to be further improved by professionals.

Author Response

Point 1: Line 22: what's meaning of active infection, please describe this issue in detail.

Done!

Point 2: Line 133: In results part, None of the swab samples tested positive for either AIV or SARS-CoV-2 viral RNA. Why was the virus only detected in the serum sample but not in the swab? The serological test might have resulted in a false positive.

Please refer to line #s 209-221 for a detailed explanation on this.

Point 3: Line 44: Which branch does the H5N1 detected in this paper belong to? And how is it different and related to the H5N1 that infected dairy cows at American dairy farms in 2023?

Please read the first two sentences in the discussion section, line # 163-166. Also, the first H5N1 infection in dairy cows in the U.S. were detected on Mar 25, 2024. 

Comments on the Quality of English Language

The language and logic of the English papers need to be further improved by professionals.

Done!

Round 2

Reviewer 1 Report

Comments and Suggestions for Authors

Unfortunately, the authors did little to address the comments, follow up comments are in italics

    1. A map showing the locations of the marshes/shelters showing the number of samples obtained from each would be helpful. The authors should at least mention the number of marshes and the number of shelters that were included
    2. Were naso-pharyngeal swabs really obtained or were they nasal swabs? Obtaining naso-pharyngeal swabs from small mammals is tricky and requires sedation. The authors response should be reflected in the manuscript and not only in the response letter.
    3. The tip of the smaller size swab is of calcium alginate which could reduce the yield of DNA and interfere with PCR. This should be discussed. The authors mention in the response that calcium alginate swabs were not used. This leaves the regular swabs that could not have been used to obtain nasal or nasopharyngeal swabs from the smaller animals sampled (chipmunks, mice, bats, etc…). The authors should provide sufficient details on how sampling was done.
    4. Were individual shelter animals sampled more than once? The response should be reflected in the manuscript.
    5. How was sample size determined or was posthoc power analysis conducted especially that swabs were negative. Lines 87-90 do not address sample size or opportunistic sampling.
    6. How were sampling sites selected? Lines 85-88 do not mention how were the sites selected.
  • A more detailed presentation of the serology data should be presented in a table ex. summarizing titers by categories (<20, 20, 40, 80, ….) and the percent with each category. The supplementary table should be moved to the main text as it is important for the readers to know the titer distribution.
  • Statistical analysis for GMT comparisons is needed for both overall and within month. Figure 2 and its legend do not have any statistical testing, only a presentation of GMTs. GMTs should be compared using a statistical test.
  • In the discussion, it is essential to better describe and add a lot more detail on how antibody detection was following virus detection in wild birds at each sampling site. Additionally, figure 2 appears to show a seasonal pattern with antibodies dropping over the summer months, this should also be discussed. Seasonal pattern is still not discussed.

Minor comments:

  • In the abstract, refrain from using abbreviations especially in the sentence showing the serological assays as it is confusing and rather spell out the assay used as indicated later in the methods ex. NP ELISA was used to screen sera and positives were further tested by VN for H5 and ELLA for N1. The abstract remains unchanged.
  • Abstract line 28: Delete ‘and recovery’ Serological findings cannot conclude that the animals had an active clinical infection that they recovered from.
  • In figure 2, it would be helpful to show the GMT value over the dots Edit the figure as some numbers and dots are overlapping

Author Response

Comments and Suggestions for Authors

Unfortunately, the authors did little to address the comments, follow up comments are in italics

We sincerely appreciate the reviewer’s time and effort in providing thoughtful and constructive feedback on our manuscript an apologize for not making an appropriate attempt to address suggestions with the first revision. As detailed below, we have addressed all comments and incorporated the suggested revisions into the manuscript to the best of our knowledge.

    1. A map showing the locations of the marshes/shelters showing the number of samples obtained from each would be helpful. The authors should at least mention the number of marshes and the number of shelters that were included.  

Number of marshes and shelters (rehab. centers) are provided in line # 230 and 232, respectively.

    1. Were naso-pharyngeal swabs really obtained or were they nasal swabs? Obtaining naso-pharyngeal swabs from small mammals is tricky and requires sedation. The authors response should be reflected in the manuscript and not only in the response letter.

We have now edited the related piece in the manuscript and pointed out that the smaller size swabs (calcium alginate-tipped) were used for small wild mammals in rehab centers (line # 240-242). We also acknowledged this as a limitation of the study in the discussion section (Line# 260-263).

The tip of the smaller size swab is of calcium alginate which could reduce the yield of DNA and interfere with PCR. This should be discussed. The authors mention in the response that calcium alginate swabs were not used. This leaves the regular swabs that could not have been used to obtain nasal or nasopharyngeal swabs from the smaller animals sampled (chipmunks, mice, bats, etc…). The authors should provide sufficient details on how sampling was done.

The materials and methods section has been edited to correctly describe the methods used in the study; small wild mammals in rehab centers were sampled using the smaller size swabs (calcium alginate-tipped) (line # 240-242) and the limitation of swab material interference has been included in the discussion section (Line# 260-263)

    1. Were individual shelter animals sampled more than once? The response should be reflected in the manuscript.

This piece is now reflected in the manuscript (line # 237-238).

    1. How was sample size determined or was posthoc power analysis conducted especially that swabs were negative. Lines 87-90 do not address sample size or opportunistic sampling. 

Sampling was done opportunistically. This information is now reflected in the manuscript (line 224-225).

    1. How were sampling sites selected? Lines 85-88 do not mention how were the sites selected.

This information is now reflected in the manuscript (line # 222-224).

  • A more detailed presentation of the serology data should be presented in a table ex. summarizing titers by categories (<20, 20, 40, 80, ….) and the percent with each category. The supplementary table should be moved to the main text as it is important for the readers to know the titer distribution.

The supplementary table is now provided in the main manuscript as Table 2 (line #s 325-328).

  • Statistical analysis for GMT comparisons is needed for both overall and within month. Figure 2 and its legend do not have any statistical testing, only a presentation of GMTs. GMTs should be compared using a statistical test.

The statistical analysis, evaluating the statistical significance of GMTs are now provided for both overall GMTs (line #s 343-344) and within month comparisons of the groups (Figure 2 and its legend). The method section is now also updated with this new piece (line # 296-300)

  • In the discussion, it is essential to better describe and add a lot more detail on how antibody detection was following virus detection in wild birds at each sampling site. Additionally, figure 2 appears to show a seasonal pattern with antibodies dropping over the summer months, this should also be discussed. Seasonal pattern is still not discussed.

 Information related to the seasonal pattern in observing GMTs are now provided in line #s 199-209.

Minor comments:

  • In the abstract, refrain from using abbreviations especially in the sentence showing the serological assays as it is confusing and rather spell out the assay used as indicated later in the methods ex. NP ELISA was used to screen sera and positives were further tested by VN for H5 and ELLA for N1. The abstract remains unchanged.

Abbreviations have been removed from the abstract

  • Abstract line 28: Delete ‘and recovery’ Serological findings cannot conclude that the animals had an active clinical infection that they recovered from.

“and recovery” is now deleted (line # 34)

  • In figure 2, it would be helpful to show the GMT value over the dots Edit the figure as some numbers and dots are overlapping

Thank you for drawing attention to the formatting errors. The figure has been updated for clarity.

Reviewer 2 Report

Comments and Suggestions for Authors

The manuscript is improved with revision. I have no further comments.

Author Response

Thank you! We sincerely appreciate the reviewer’s time and effort in providing thoughtful and constructive feedback on our manuscript. 

Reviewer 3 Report

Comments and Suggestions for Authors

The author's revisions and responses to questions are inappropriate, and there are also flaws in the detection method of the article. Therefore, it is not recommended to accept this article.

Author Response

We sincerely appreciate the reviewer’s time and effort in providing thoughtful and constructive feedback on our manuscript an apologize for not making an appropriate attempt to address suggestions with the first revision. As detailed below, we have addressed all comments and incorporated the suggested revisions into the manuscript to the best of our knowledge.

Point 1: Line 22: what's meaning of active infection, please describe this issue in detail.

Active infection is now defined in the abstract. Please refer to the line #s 23-24.

Point 2: Line 133: In results part, None of the swab samples tested positive for either AIV or SARS-CoV-2 viral RNA. Why was the virus only detected in the serum sample but not in the swab? The serological test might have resulted in a false positive.

Detection of active virus (viral RNA) from the swab samples were not expected in the seroconverted animals, as the immune response and ability to detect antibodies post-infection is delayed post-exposure. Because this surveillance effort was done opportunistically the animals in this study were sampled primarily during seasons in which wild, migratory waterfowl were not moving south into the geographical location; and thus the estimated prevalence of HPAI H5N1 is low in the waterfowl population. This is a significant limitation of the study, however the high seroprevalence indicates the animals were exposed when wild waterfowl were indeed shedding virus in the habitats. This limitation is  addressed in line # 109-209.

Point 3: Line 44: Which branch does the H5N1 detected in this paper belong to? And how is it different and related to the H5N1 that infected dairy cows at American dairy farms in 2023?

The H5N1 seroconversion detected in this study belongs to the Goose/Guandong (Gs/GD) lineage HPAI H5N1 viruses belonging to clade 2.3.4.4b, spread from Europe into North American through migratory birds in December 2021. This is the same clade that has affected multiple animal species, including American dairy cows. This information are provided in line #s 187-193.  Unfortunately, the serology testing methodologies used in this study are not sensitive enough to discern antigenic differences between sub-lineages (ex. B1, D1.1, etc.)

Comments on the Quality of English Language

The language and logic of the English papers need to be further improved by professionals.

Editing to improve the readability of this manuscript has been completed by senior professors at The Ohio State University.

Round 3

Reviewer 3 Report

Comments and Suggestions for Authors

I am  satisfied with the revision of this paper and the author's response. It has basically reached the standards of the journal and I recommend its acceptance.